# FE-GNN: Feature Enhanced Graph Neural Networks for Account Classification in Ethereum

## Abstract

Since the birth of the blockchain cryptocurrency trading platform represented by Bitcoin, cryptocurrencies based on blockchain technology have gained widespread attention and accumulated a large amount of transaction data. The analysis of cryptocurrency transactions has become an important research direction with social and economic value, and an important area of blockchain scientific research. Identifying the identity of different cryptocurrency addresses and understanding their behavior is the core challenge to achieve cryptocurrency transaction analysis, otherwise it is difficult to understand blockchain datasets and analyze them with meaningful results. To this end, this paper proposes a blockchain address identity identification method called **F**eature **E**nhanced **G**raph **N**eural **N**etworks (FE-GNN). Specifically, a transaction graph is constructed based on the collected transaction data, and graph learning techniques based on graph convolutional networks and graph attention networks are used to infer the blockchain address identity. Experimental results show that the FE-GNN algorithm outperforms previous algorithms.

## 1 Introduction

Cryptocurrency is a digital currency built on blockchain technology that enables blockchain transactions over the Internet without a trusted third party. However, the anonymity of cryptocurrencies allows the real identity of transaction users to be concealed, leading to bitcoin being used by some unscrupulous elements in various illegal activities, for example, using cryptocurrencies for money laundering (Sun et al., 2022), fraud (Jung et al., 2019), theft of funds (Lazarenko & Avdoshin, 2018), dark web market transactions (Kanemura et al., 2019), terrorist financing (Nguyen, 2016), which is broad in scope and may involve any transaction involving the transfer of property. As well as various traditional crimes such as relationship scams and pyramid schemes (Fan et al., 2021; Chen et al., 2022a). Finally there are various counterfeit frauds against the blockchain system, such as impersonating exchanges and wallets (Andryukhin, 2019), issuing fake ERC20 tokens on Ethereum (Gao et al., 2020) and USDT (Chen et al., 2022b) for fraud.

Compared with traditional financial systems, the unique characteristics of cryptocurrencies, such as address anonymization and transaction decentralization, make their transactions have strong anti-traceability, which also leads to many challenges for the identification mechanism of cryptocurrency transaction addresses.Existing cryptocurrency identification methods mainly obtain the representation of nodes through graph neural networks or graph representation learning methods and perform node classification to achieve the identification of cryptocurrency transaction addresses. However, there are still two legacy problems: **(1)Lack of effective node representation.** Existing methods mostly dichotomize nodes for phishing, fraud, and other types of nodes, without considering other types of nodes in cryptocurrencies, such as miners, exchanges, and ICO wallets. **(2)Ignoring node types and transaction types.** The existing methods ignore the difference between external accounts and contract accounts in cryptocurrencies, and also ignore transaction types, without considering transaction types such as transferring, creating contracts, and invoking contracts in transactions.

To solve the above problem, this paper proposes **F**eature **E**nhanced **G**raph **N**eural **N**etworks (FE-GNN) to enhance cryptocurrency node classification detection by learning stronger node representa-

Figure 1: The proposed Ethereum account identification method framework includes three components, namely (a) Convolutional Layer, (b) Self-attention Layer and (c) Enhancing Framework.

tions. By analyzing a large number of labeled accounts in cryptocurrencies, transaction features are extracted, accounts are abstracted as nodes, and transfers between accounts are abstracted as edges. Then a new meta-path graph structure is generated based on the above transaction network, and a more efficient graph convolution is performed on the new graph to learn a stronger node representation. Finally, node classification is performed to achieve recognition of cryptocurrency transaction addresses.

The main contributions of this article are summarized as follows.

- This paper proposes a node feature collection strategy. By analyzing the transaction data of each account in the cryptocurrency, it can comprehensively and accurately describe the transaction behavior of nodes, making up for the shortcomings of only focusing on transaction records.

- This paper collects and labels 2286 labeled nodes (specifically: Exchange, ICO Wallets, Investment, Miner, Phish, Ponzi, Token Contract). And retrieve the relevant transaction and block data according to the labeled node, and collect an Ethereum transaction dataset containing 1,124,130 nodes and 3,752,659 edges.

- This paper proposes a method for cryptocurrency identification. This method proposes two feature enhancement components, convolutional layer and self-attention layer, to solve the Ethereum account classification problem. With these two components, more efficient graph learning is performed on the graph, resulting in stronger node representations.

- Extensive experiments are conducted on the collected dataset of Ethereum transactions, and the results show that the algorithm proposed in this paper outperforms the state-of-the-art methods in several metrics.

## 2 BACKGROUND

### 2.1 BLOCKCHAIN ACCOUNT CLASSIFICATION METHOD

In recent years, as cryptocurrencies continue to mature, the price of cryptocurrencies such as bitcoin and Ethereum has climbed significantly, and the number of users continues to increase. Meanwhile, as an emerging interdisciplinary research field, the research on the identification of blockchain cryptocurrency transaction addresses has attracted the attention of a large number of scholars. Some research has already yielded results, such as smart contract Ponzi scheme detection, money laundering detection, coin mix detection, fraud detection and phishing detection.

Chen et al. (2019) proposed a classification model for detecting smart Ponzi schemes by extracting two kinds of features from the transaction records and the operation code of smart contracts. Bartoletti et al. (2018) approach was similar to Chen et al., except that they used data mining techniques to identify bitcoin-related scams. Henderson et al. (2012) proposed a method of using K-means and Role Extraction (RolX) to be able to identify bitcoin users who are laundering money on the Bitcoin network, providing a visual depiction of the interaction of money laundering accounts, showing how bitcoins are repeatedly segmented and directed to new addresses. They use a variety of machine learning algorithms to conduct experiments on the data sets they collect, and evaluate the results of the experiments to verify the effectiveness of the method.

In recent years, Deep learning based graph representation learning methods are also widely used for blockchain node classification. Weber et al. (2019) proposed a bitcoin antimoney laundering method using graph convolutional networks. Wu et al. (2021) combined the transaction network structure to construct feature data and used the semi-supervised machine learning algorithm PU learning to build a hybrid coin recognition model. Wang et al. (2022) proposed a heterogeneous network representation learning method to mine implicit information inside Ethereum transactions. Liu et al. (2022b) proposed an identity inference approach by graph learning for Ethereum and other similar DApp platform blockchains.

The data in the blockchain contains multiple information with high dimensionality, and graph embedding can be an excellent solution to this problem. Yuan et al. (2020a) used node2vec for phishing node classification. Wu et al. (2022) proposed a method to detect phishing scams by digging through the transaction records of Ethereum. The method extracts address features by proposing a new network embedding algorithm trans2vec, and then uses One-Class SVM to classify Ethereum nodes into ordinary nodes and phishing nodes. Yuan et al. (2020b) used an improved Graph2Vec based implementation for classification prediction of the constructed transaction subgraphs. Lin et al. (2020) analyzed Ethereum transactions by a time-weighted multiple graph embedding method, which models the Ethereum transaction network as Temporal Weighted Multidigraph. Blockchain network analysis based on graph embedding emphasizes transaction information and ignores the attributes of illegal nodes, which reduces the prediction accuracy.

# 3 METHOD INTRODUCTION

## 3.1 ETHEREUM ACCOUNT DATASET

### 3.1.1 DATA COLLECTION

In this work, node tagging information is acquired from the Etherscan[1] tag word cloud module. Subsequently, the Etherscan application programming interface (API) is employed to retrieve all transaction data associated with the tagged nodes.

This API supports obtaining the latest 10,000 normal and internal transactions for contract accounts (CA) and externally owned accounts (EOA). By configuring the API parameters with the address where node label information is collected, transaction records for all accounts can be extracted, thus providing the necessary transaction data for this research.

### 3.1.2 ACCOUNT FEATURES EXTRACTION

Due to the anonymity of the blockchain platform, the blockchain accounts themselves do not contain any attribute information. In order to better describe the behavior of different accounts and achieve excellent classification. Based on the transaction history of the accounts, this paper considers the number, value and frequency of transactions and other easily calculable data. Thirty account features are extracted, as shown in Table 1. These features can further reveal the correlation between trading behavior and accounts to discover the variability of trading patterns among different accounts. Some of these features are described as follows.

The number of transactions sent (NTS): the number of transactions sent from an account, $\text{NTS}_i$ represents the number of transactions sent from account $i$.

---

[1]Etherscan, https://etherscan.io/labelcloud

Table 1: Complete list of the 30 extracted features.

| | Extracted Feature | Description | Data Type |
|---|---|---|---|
| **1** | NTS | The number of transactions sent | Integer |
| **2** | max_VTS | The maximum value of transactions sent | Double |
| **3** | min_VTS | The minimum value of transactions sent | Double |
| **4** | TVS | The total value of transactions sent | Double |
| **5** | AVS | The average value of transactions sent | Double |
| **6** | avg_TIS | The average time interval between transactions sent | Integer |
| **7** | NTR | The number of transactions received | Integer |
| **8** | max_VTR | The maximum value of transactions received | Double |
| **9** | min_VTR | The minimum value of transactions received | Double |
| **10** | TVR | The total value of transactions received | Double |
| **11** | AVR | The average value of transactions received | Double |
| **12** | avg_TIR | The average time interval between transactions received | Integer |
| **13** | TETF | The total ethereum transaction fee | Double |
| **14** | AETF | The average ethereum transaction fee | Double |
| **15** | TDFL | The time difference between the first and last transaction | Integer |
| **16** | USA | The unique send address | Integer |
| **17** | URA | The unique receive address | Integer |
| **18** | TEB | The total ethereum balance after the transaction | Double |
| **19** | ERC20_NTS | The number of ERC20 token transactions sent | Integer |
| **20** | ERC20_max_VTS | The maximum value of ERC20 tokens transactions sent | Double |
| **21** | ERC20_min_VTS | The minimum value of ERC20 tokens transactions sent | Double |
| **22** | ERC20_TVS | The total value of ERC20 token transactions sent | Double |
| **23** | ERC20_AVS | The average value of ERC20 token transactions sent | Double |
| **24** | ERC20_NTR | The number of ERC20 token transactions received | Integer |
| **25** | ERC20_max_VTR | The maximum value of ERC20 tokens transactions received | Double |
| **26** | ERC20_min_VTR | The minimum value of ERC20 tokens transactions received | Double |
| **27** | ERC20_TVR | The total value of ERC20 token transactions received | Double |
| **28** | ERC20_AVR | The average value of ERC20 token transactions received | Double |
| **29** | ERC20_USA | The unique ERC20 token send address | Integer |
| **30** | ERC20_URA | The unique ERC20 token receive address | Integer |

The total value of transactions sent (VTS): The sum of the transaction values sent by the account, $VTS_i$ represents the sum of the transaction values sent from account $i$.

The average value of transactions sent (AVS): represents the average value of transactions sent by an account, which can be calculated from the current account NTS and VTS, calculated as:

$$SAV_i = \frac{STV_i}{NTS_i} \tag{1}$$

where $VTS_i$ represents the average value of transactions sent from account $i$.

The maximum value of transactions sent (max_VTS) and the minimum value of transactions sent (min_VTS), which represent the maximum and minimum time interval between two transactions for a given account, respectively. $T_{i,k}$ denotes the timestamp of the $k$-th transaction sent by account $i$. The max_VTS and min_VTS are calculated as:

$$\max\_VTS_i = \max_k \left( |T_{i,k+1} - T_{i,k}| \right) \tag{2}$$

$$\min\_VTS_i = \min_k \left( |T_{i,k+1} - T_{i,k}| \right) \tag{3}$$

The average time interval between transactions sent (avg_TIS): represents the average time interval of sending transactions for an account, which can be calculated from the time interval of each transaction and NTS. $avg\_TIS_i$ represents the average time interval of sending for account $i$, $k$ is the total number of transactions for account $i$ and is calculated as:

$$avg\_TIS_i = \frac{\sum_{j=1}^{k} T_{i,j+1} - T_{i,j}}{NTS_i} \tag{4}$$

The number of transactions received, maximum value of transactions received, minimum value of transactions received, total value of transactions received, average value of transactions received, average time interval between transactions received features are calculated in a manner similar to the features of sending transaction accounts, and are calculated as in Eq.(1) to (4).

The total ethereum transaction fee (TETF): the sum of transaction fees for each account, which can be calculated from the price of gas and gas used in the transaction. $k$ is the number of transactions for the $i$-th account. $PG_{i,j}$ and $GU_{i,j}$ represent the price of gas and gas used in the $j$-th transaction for the $i$-th account, respectively. And uniformly convert Wei to Ether, calculated as:

$$k = NTS_i + NTR_i \tag{5}$$

$$TETF_i = \sum_{j=1}^{k} (GU_{i,j} \times PG_{i,j}) \times 10^{-18} \tag{6}$$

The average ethereum transaction fee (AETF): The average of transaction fees for an account, which can be obtained from the TETF and the number of transactions, calculated as:

$$AETF_i = \frac{TETF_i}{k} \tag{7}$$

The feature numbers 19-30 are calculated as in Eq.(1) to (7).

### 3.1.3 IDENTITY CATEGORIZATION

For effective identification, some common Ethereum account identity types are selected in this paper. Table 2 shows a breakdown of the Ethereum accounts used during the experiments. The appendix A contains detailed descriptions of the account types.

Table 2: Typical Account Identities

| Identity | Type | Number |
|---|---|---|
| Exchange | EOA/CA | 518 |
| ICO Wallets | EOA/CA | 163 |
| Investment | CA | 74 |
| Miner | EOA/CA | 192 |
| Phish | EOA/CA | 664 |
| Ponzi | CA | 48 |
| Token Contract | CA | 627 |

### 3.2 FRAMEWORK

The framework of FE-GNN is shown in Fig. 1. As shown in the Fig. 1, the method consists of three parts, namely convolutional layer, self-attentive layer, and enhancing framework. **(1) Convolutional layer.** In a transaction network, transaction types are complex. To explore the impact of different transaction types on node representation. A new graph structure is generated and multiple candidate adjacency matrices are used to find a new graph structure for a more efficient graph convolution. **(2) Self-Attention Layer.** The adjacency matrix constructed based on convolutional layers defines a transformed isomorphic network that utilizes a self-attention mechanism to compute the hidden representation of each node by paying attention to its neighbors. **(3) Enhanced Framework.** The Enhanced framework repeatedly stacks multiple convolutional layers and self-attention layers, gradually enhancing node features in this way.

### 3.2.1 CONVOLUTIONAL LAYER

Previous work dealing with heterogeneous graphs required manually defining meta-paths, generating adjacency matrices from meta-paths, and executing graph neural networks. However, there is

no meta-path related experiments on the Ethereum dataset for reference. Therefore, a method is proposed to learn the meta-path graph of an Ethereum dataset and perform GCN operations on the learned meta-path graph. The specific process is shown in the Fig. 1(a).

Based on the above idea a $l$-layer meta-path adjacency matrix calculation method is designed, specifically, a convolution kernel is formed using softmax to convolve the adjacency matrix, and the convolution results in a similar weighted summation of the adjacency matrix, which is calculated as follows

$$A^{(l)} = \text{conv}_{1 \times 1} \left( \mathbb{A}; \text{softmax} \left( \phi^{(k)} \right) \right) \tag{8}$$

$$= \sum_{t=1}^{|\mathcal{T}_e|} \alpha_t^{(k)} A_t \tag{9}$$

where $\alpha^{(k)} = \text{softmax} \left( \phi^{(k)} \right)$, $\phi^{(k)} \in R^{1 \times 1 \times |R|}$ is the parameter of $1 \times 1$ convolution, $|R|$ is the number of edge type.

The output is then multiplied with the output matrix of the previous layer and the output matrix is normalized, which is calculated as follows

$$A^l = \left( \hat{D}^{(l)} \right)^{-1} A^{(l-1)} A^{(l)} \tag{10}$$

where $\hat{D}^{(l)}$ is the degree matrix after multiplying the two matrices.

Next, the convolutional structure is used to learn different node representations. Specifically, the constructed meta-path adjacency matrix $A$ is applied to the GCN, and the node representations are extracted end-to-end using the GCN. The proposed GCN architecture with sub-layers following the propagation rules:

$$H^{(l+1)} = \sigma \left( D^{-\frac{1}{2}} A^l D^{-\frac{1}{2}} H^{(l)} W^{(l)} \right) \tag{11}$$

where $D$ is a diagonal matrix with $D_{ii} = \sum_j A^l_{ij}$, and $W^{(l)}$ is a layer-specific trainable weight matrix. $\sigma(\cdot)$ is an activation function such as ReLU or Sigmoid. $H^{(0)} = X$ is the input node features, and $H^{(l)} \in \mathbb{R}^{N \times d}$ the output node features of the $l^{th}$ layer.

Finally, the representations of multiple nodes are concatenated

$$Z = \|_{i=1}^C \sigma \left( D^{-\frac{1}{2}} A^l D^{-\frac{1}{2}} H^{(l)} W^{(l)} \right) \tag{12}$$

where $\|$ is the concatenation operator, $C$ denotes the number of layer.

### 3.2.2 SELF-ATTENTION LAYER

As shown in Fig. 1(b), use the method mentioned in the previous section to construct an adjacency matrix, defined as $A^{(l)} \in \mathbb{R}^{N \times N}$ in Fig. 1(b), where $N$ stands for the total number of nodes, then leverages self-attention to compute the representation of each node by paying attention to its neighbors.

First, self-attention to the target node is achieved by designing an attention mechanism. The attention mechanism is denoted as $a : \mathbb{R}^{d' \times d'} \to \mathbb{R}$ , where $d'$ is the output dimension of self-attention Layer, $a$ is a single feed-forward layer with non-linearity. $a$ takes the linearly transformed representations of two nodes as input and output an attention coefficient:

$$e_{ij} = a \left( \boldsymbol{W} \boldsymbol{v}_{li}, \boldsymbol{W} \boldsymbol{v}_{lj} \right) \tag{13}$$

$$= \sigma \left( \boldsymbol{a}^{\mathrm{T}} \left[ \boldsymbol{W} \boldsymbol{v}_{li} \| \boldsymbol{W} \boldsymbol{v}_{lj} \right] \right) \tag{14}$$

where $\boldsymbol{v}_{li} \in \mathbb{R}^d$ denotes the input representation of node $v_{li}$, $\boldsymbol{v}_{lj} \in \mathbb{R}^d$ denotes the input representation of node $v_{lj}$. $\boldsymbol{W} \in \mathbb{R}^{d' \times d}$ is a weight matrix. $\boldsymbol{a} \in \mathbb{R}^{2d'}$ is the linear transformation weight matrix applied over each node. $\sigma(\cdot)$ denotes the nonlinear function, and $\|$ stands for the concatenation operation. where $\boldsymbol{W}$ and $\boldsymbol{a}$ are shared among all node pairs. The attention coefficient $e_{lj}$ indicates the importance of $v_{lj}$ 's representation to $v_{li}$.

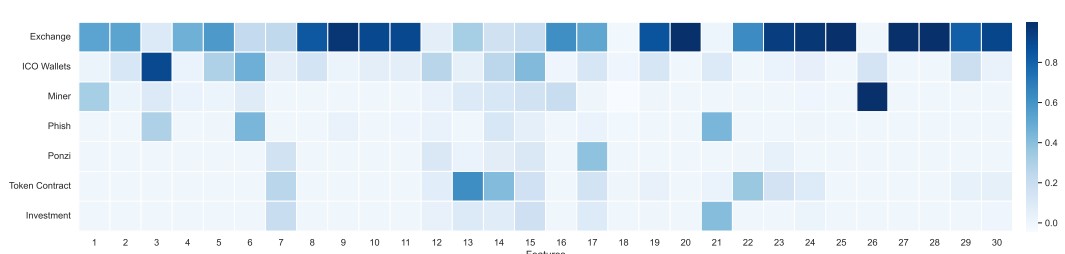

Figure 2: Classification results (%) for different combination patterns.

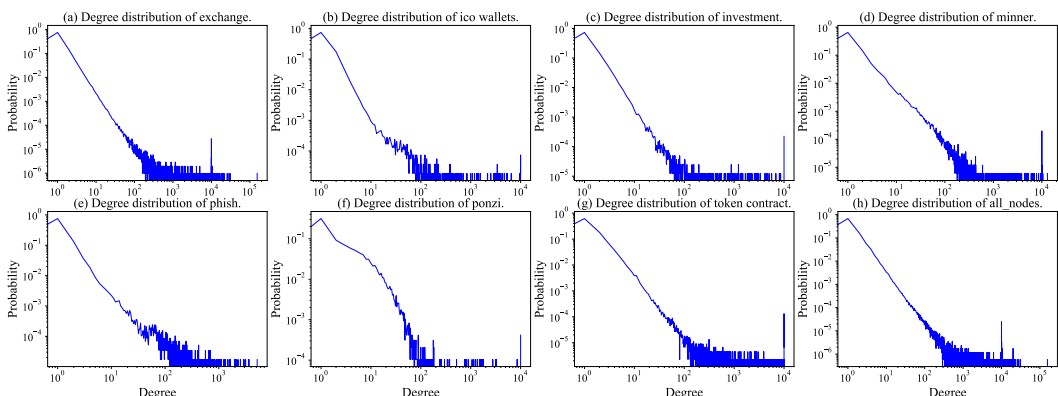

Figure 3: Degree distribution of all nodes

To make the attention coefficients between different nodes easy to compare, the attention coefficients of all for nodes are normalized using the softmax function:

$$\alpha_{ij} = \mathrm{softmax}\left(e_{ij}\right) \tag{15}$$

$$= \frac{\exp\left(\sigma\left(\boldsymbol{a}^T\left[\boldsymbol{W}\boldsymbol{v}_{li}\|\boldsymbol{W}\boldsymbol{v}_{lj}\right]\right)\right)}{\sum_{n\in\mathcal{N}_i}\exp\left(\sigma\left(\boldsymbol{a}^T\left[\boldsymbol{W}\boldsymbol{v}_{li}\|\boldsymbol{W}\boldsymbol{v}_{ln}\right]\right)\right)} \tag{16}$$

where $\mathcal{N}_i$ is a set of $v_{li}$ 's first-order semantic structure-based neighbors according to $\boldsymbol{A}'$. $\alpha_{ij}$ is asymmetric. The output representation of $v_{li}$ can then be computed by paying attention to its neighbors using the normalized attention coefficients:

$$\boldsymbol{v}'_{li} = \sigma\left(\sum_{v_{lj}\in\mathcal{N}_i}\alpha_{ij}\boldsymbol{W}\boldsymbol{v}_{lj}\right) \tag{17}$$

Next, In order to stabilize the learning process of self-attention, this paper uses a multi-headed attention mechanism. Specifically, $H$-independent attention mechanisms are trained and connect their outputs as the final representation.

$$\boldsymbol{v}'_{li} = \|_{h=1}^H \sigma\left(\sum_{v_{lj}\in\mathcal{N}_i}\alpha_{ij}^h\boldsymbol{W}^h\boldsymbol{v}_{lj}\right) \tag{18}$$

where $\alpha_{ij}^h$ stands for the head-wise normalized attention coefficients, and $\boldsymbol{W}^h$ stands for the head-wise linear transformation matrix. In this paper, the output dimension of each head is set to $d' = d/H$, such that the output dimension of self-attention layer is equal to its input dimension.

Table 3: Three training-validation-test set (%) division methods.

| Dataset | Train set | Validation set | Test set |
|---------|-----------|----------------|----------|
| $D_1$ | 30% | 30% | 40% |
| $D_2$ | 60% | 20% | 20% |
| $D_3$ | 80% | 10% | 10% |

### 3.2.3 ENHANCED FRAMEWORK

As shown in Fig. 1(c), the augmentation framework stacks multiple Convolutional Layers and Self-Attention Layers on top of each other. It constructs a method to enhance the node features of the input by Convolutional Layer and Self-Attention Layer.

## 4 EXPERIMENT

In this section, an experimental evaluation is conducted to investigate the effectiveness of the FE-GNN proposed in this paper for the account classification task in the collected Ethereum transaction dataset.

### 4.1 DATASET AND EVALUATION CRITERIA

#### 4.1.1 DATA COLLECTION

Ethereum is currently the largest blockchain smart contract blockchain encryption platform, and there is a rich tag library. Therefore, it is possible to classify Ethereum accounts and identify the different accounts. The Ethereum dataset is constructed using the method proposed in Section III-A and the performance of the FE-GNN proposed in this paper is evaluated using this dataset. The details of the marked nodes are shown in Table 2. The dataset includes a total of 8 common account labels such as exchange, ICO wallets, investment, miners, Phish, ponzi and token contract.

The final constructed Ethereum dataset includes 1,124,130 nodes and 3,752,659 edges. To effectively evaluate FE-GNN, the initial data set is divided according to the scale shown in Table 3, as referenced in Liu et al. (2022a). The training sets in $D_1$, $D_2$, and $D_3$ contain 30%, 60%, and 80% of randomly selected labeled nodes, respectively. During model training, validation and testing, only the classification performance of 2286 labeled nodes is considered.

#### 4.1.2 COMPARISON METHODS

The baseline methods were compared by analyzing similar work. The FE-GNN method is compared with several methods, including (1) feature-based methods that consider only node attributes (i.e., Logistic Regression (Wright, 1995), Random Forest (Ho, 1995)); (2) Random walk-based network embedding methods (i.e., DeepWalk (Perozzi et al., 2014), Node2Vec (Grover & Leskovec, 2016)); (3) Some popular deep learning network-based despicable methods (i.e., GCN (Kipf & Welling, 2016), GAT (Veličković et al., 2017), BI-FedGNN (Gao et al., 2024)); (4) Some popular Ethereum phishing node detection methods (i.e., $T^2A2vec$ (Wang et al., 2023), HNRL (Wang et al., 2022)).

The parameters of the above methods all adopt the optimal parameter settings in the paper. In each experiment, the dataset was randomly divided according to the proportion in Table 3, each method was run 10 times, and the results were averaged.

### 4.2 ETHEREUM ACCOUNT CLASSIFICATION RESULTS

This paper evaluates the performance of different methods on the Ethereum identity recognition task, and the results are shown in Table 4. From this, the following conclusions can be drawn:

Table 4: The classification results (%) over the methods

| Method | Dataset | $D_1$ | | | | $D_2$ | | | | $D_3$ | | | |
|---|---|---|---|---|---|---|---|---|---|---|---|---|---|
| | Metric | Pre.[1] | Recall | Mi-F1 | Ma-F1 | Pre.[1] | Recall | Mi-F1 | Ma-F1 | Pre.[1] | Recall | Mi-F1 | Ma-F1 |
| Feature-based | LR[2] | 37.28 | 33.76 | 57.06 | 31.98 | 39.38 | 32.84 | 56.61 | 32.09 | 38.69 | 34.66 | 56.12 | 33.85 |
| | RF[3] | 62.12 | 59.60 | 73.58 | 59.83 | 62.47 | 58.94 | 73.51 | 59.51 | 69.61 | 64.79 | 75.72 | 66.26 |
| Random walk | DeepWalk | 43.17 | 40.76 | 55.99 | 41.45 | 45.83 | 42.98 | 66.15 | 43.36 | 40.97 | 42.43 | 60.49 | 41.30 |
| | Node2Vec | 45.45 | 45.92 | 52.77 | 45.54 | 44.9 | 46.58 | 55.41 | 45.53 | 49.64 | 50.46 | 56.95 | 49.95 |
| Deep learning | GCN | 51.81 | 41.56 | 58.32 | 43.17 | 56.21 | 40.91 | 59.38 | 42.82 | 57.78 | 42.77 | 60.52 | 45.34 |
| | GAT | 76.91 | 73.62 | 79.30 | **74.48** | 81.82 | **79.56** | 83.31 | **79.52** | 78.69 | 76.37 | 80.13 | 76.32 |
| | BI-FedGNN | 72.45 | 71.26 | 76.83 | 72.22 | 74.98 | 73.75 | 77.03 | 72.46 | 77.36 | 75.36 | 78.93 | 75.26 |
| Ethereum methods | $T^2A2vec$ | 52.64 | 41.95 | 58.20 | 43.80 | 56.60 | 45.88 | 47.19 | 62.47 | 56.16 | 47.00 | 63.36 | 48.33 |
| | HNRL | 60.39 | 56.34 | 71.25 | 55.22 | 71.15 | 68.06 | 78.36 | 67.57 | 75.72 | 73.74 | 81.31 | 75.53 |
| | FE-GNN | **77.94** | **76.98** | **80.31** | 72.23 | **82.62** | 78.97 | **84.54** | 77.02 | **83.26** | **80.31** | **87.63** | **79.17** |

[1] Precision.
[2] Logistic Regression.
[3] Random Forest.

(1) FE-GNN achieves significant advantages under different evaluation metrics, with 83.26% precision and 85.92% recall for FE-GNN, 81.62% for Mi-F1, and 80.53% for Ma-F1. The second best method is GAT, whose method ranks second in most cases. The next best method is the deep learning based method, but there is a large gap between different deep learning methods, for example, GCN has a performance gap of nearly 25% for $D_1$ dataset. The difference in performance between the two random walk based methods is not much on average around 5%. There are also two extremes in the feature-based methods, and the worst performance is the logistic regression method, which has an accuracy of only about 38.69%. The Mi-F1 of the random forest algorithm is 75.72%.

(2) The performance of all algorithms keeps improving as the proportion of training set keeps increasing, with the random forest algorithm showing the largest performance improvement. The improvement rate of the method proposed in this paper follows closely. The deep learning-based method and the random walk based method are next.

(3) Compared with the feature-based methods, the four evaluation metrics of FE-GNN outperform them by 10%-40%. Among all the compared methods, only logistic regression has the worst performance. The reason for this situation may be the small number of node features collected in this paper, which limits the effect of logistic regression. But the randomized deep forest algorithm achieved good results, which verifies the effectiveness of the node features collected in this paper.

(4) For methods based on random wandering achieved generally poor results, Node2vec method performed the best. This is mainly because this method ignores the transaction features between nodes and cannot learn a more effective node representation.

(5) For deep learning-based methods, there is a large gap between different methods. Some methods perform poorly. For example, there is a general gap of 10%-25% between GCN and GAT and BI-FedGNN.

(6) Some of the Ethereum node classification methods reproduced in this paper have a strong competitive advantage. Among them, $T^2A2vec$ is an improvement of node2vec. $T^2A2vec$ improves the metrics by 5%-10% compared with node2vec by considering two transaction characteristics, namely transaction time and transaction amount. HNRL is an Ethereum phishing node detection method using heterogeneous graph representation learning method, which is only 6% different from FE-GNN Mi-F1 in the $D_2$ case. Although the overall performance of these methods is poor, they achieve good results for phishing node identification, with metrics exceeding 80% for both algorithms.

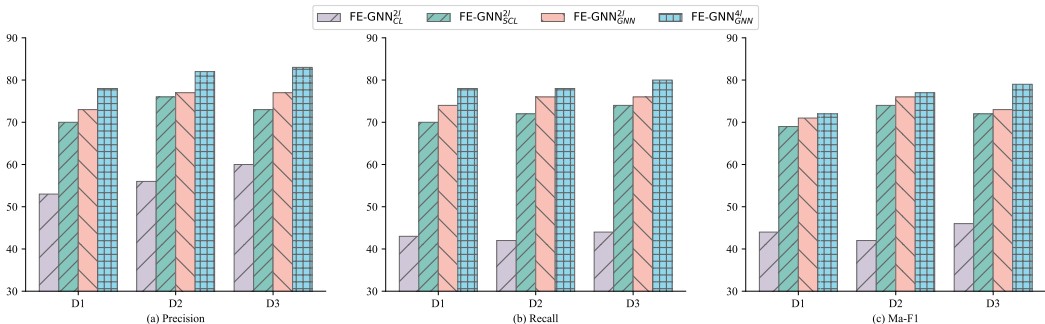

Figure 4: Classification results (%) for different combination patterns.

### 4.3 PARAMETER SENSITIVITY ANALYSIS

This paper evaluates the effect of the number of convolutional and self-attentive layers on the classification performance. Experiments were conducted on four architectures, namely $\text{FE-GNN}_{CL}^{2l}$, $\text{FE-GNN}_{SCL}^{2l}$, $\text{FE-GNN}_{GNN}^{2l}$, and $\text{FE-GNN}_{GNN}^{4l}$. Where $\text{FE-GNN}_{CL}^{2l}$ means that it contains only two convolutional layers. The $\text{FE-GNN}_{SCL}^{2l}$ representation contains only two self-attention layers. The $\text{FE-GNN}_{GNN}^{2l}$ representation consists of one convolutional layer and one self-attention layer. The $\text{FE-GNN}_{GNN}^{4l}$ representation consists of two convolutional layers and two self-attention layers.

As shown in Fig. 4, this paper evaluates their classification performance. It can be seen from the table that better classification results are achieved when convolutional layers and self-attention layers are included than when only one of them is included. Experimental results show that both convolutional and self-attentive layers can improve the classification performance, and when both are combined, better classification results are achieved. It can be found that the performance of the model becomes more stable as the number of layers increases.

## 5 CONCLUSION AND FUTURE WORK

In this paper, a Feature Enhanced Graph Neural Networks (FE-GNN) is proposed to handle the task of account classification in Ethereum. Through the analysis of Ethereum transaction data, this paper designs a node feature collection strategy, which can fully and accurately describe the transaction behavior of nodes. FE-GNN proposes two feature enhancement components, convolutional layer and self-attention layer, to solve the Ethereum account classification problem. With these two components, more efficient graph learning is performed on the graph, resulting in stronger node representations. With node representations obtained from node features and graph learning, the performance of Ethereum account classification detection is improved.Extensive experiments show that FE-GNN outperforms and outperforms the state-of-the-art algorithms in terms of performance and utility.

In future work, the method is considered for use in blockchain identity browsers, and the identification tags are stored in a library of address tags, which can alert and suggest to asset associates that the transfer may be risky and should be guarded against once it is associated with a tagged illegal address. And consider extending FE-GNN to dynamic blockchain transaction networks that include temporal information.

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

# A PRELIMINARIES

## A.1 ETHEREUM TRANSACTION GRAPH

Constructing an Ethereum transaction graph as a heterogeneous graph $G = (V, E, R)$, where $V = \{v_1, \cdots, v_n\}$ is the set of nodes, $E$ is the set of edges and $R$ is the set of edge types. The total number of accounts is $N = |V|$. Each node $v \in V$ represents contract accounts or externally owned accounts. The edge types $R$ include transfer, invoke contract, and create contract, respectively. Each node $v_i$ is associated with a feature vector $\overrightarrow{\mathbf{x}}_i \in \mathbb{R}^{d_f}$ where $d_f$ is the dimension of the feature vector. The input feature vectors of each node are concatenated into feature matrix $\mathbf{X} \in \mathbb{R}^{n \times d_f}$, where the $i$-th row is $\overrightarrow{\mathbf{x}}_i$. Each edge $e \in E$ is associated with an edge type $\phi(e) \in R$. The heterogeneous graph can be represented by a set of adjacency matrices $\{A_k\}_{k=1}^{K}$ where $K = |R|$, and $A_k \in R^{N \times N}$ is an adjacency matrix where $A_k[i, j]$ is non-zero when there is a $k$-th type edge from $j$ to $i$ . More compactly, it can be written as a tensor $\mathbb{A} \in R^{N \times N \times K}$.

## A.2 GRAPH REPRESENTATION LEARNING

Graphs are ubiquitous in the real world, covering applications ranging from social networks(Hamilton et al., 2017), recommender systems(Lu et al., 2020), knowledge graphs(Wang et al., 2014), transportation networks(Zhao et al., 2019), and drug discovery(Sun et al., 2020). Graph representation learning has been shown to be effective in many downstream tasks such as node classification(Tang et al., 2016; Zhou et al., 2007), link prediction(Singh & Gordon, 2008), graph classification(Lee et al., 2018; Wu et al., 2017) and clustering(Wang et al., 2019). Graph representation learning is attracting the attention of researchers and practitioners, becoming a research hotspot of data mining, and a large number of research results are emerging. Graph representation learning (i.e. graph embedding or network embedding) methods can be grouped into three categories: based on Factorization, Random Walk and Deep Learning.

Factorization-based graph representation learning methods (Wang et al., 2017; Yang et al., 2015; Zhang et al., 2016b; Tu et al., 2016; Zhang et al., 2016a) are early research approaches. There are two main decompositions of factorization-based graph representation learning methods, which are graph Laplacian feature graph decomposition and vertex proximity matrix decomposition.

Random walk-based graph representation learning methods (Perozzi et al., 2014; Grover & Leskovec, 2016; Ribeiro et al., 2017; Shi et al., 2018) use a flexible and random vertex similarity metric, resulting in excellent performance in many scenarios. Random walk-based methods are broadly classified into two categories, random walk methods for homogeneous graphs and embedding methods for heterogeneous graphs.

Deep learning-based graph representation learning methods (Wang et al., 2016; Li et al., 2021; Cao et al., 2016; Kipf & Welling, 2016; Hamilton et al., 2017; Veličković et al., 2017) apply deep learning to the entire graph (or adjacency matrix), and its popular deep learning models include two, autoencoders and deep neural networks.

## A.3 IDENTITY CATEGORIZATION

**Exchanges.** Similar to stock exchanges where stocks are bought and sold, a blockchain exchange is a website platform where digital currencies are bought and sold for trading. It allows traders to buy and sell cryptocurrencies using fiat currency or other cryptocurrencies. Exchanges account for a large portion of blockchain digital currency trading, and some of the more popular exchanges include Huobi, Binance, Bitfinex, Kraken, Bithumb, and others. These trading platforms generally only provide functions such as top-up, transfer and withdrawal, which means that they will only tell you the address of your wallet receipt, and the wallet key, Keystore and helper words are generally not provided. Authentication is done through login username, password, verification email, cell phone, etc.

**Miner.** Mining is the process of using computer hardware to calculate, record and verify information in a digital record known as a blockchain. Miners solve mathematical puzzles by mining to gain the right to create new blocks and the reward for the blocks that come out, so called because it works much like mineral mining. Currently, the most common way is through the proof-of-work (PoW)

consensus mechanism, where the first computer to solve a complex mathematical problem is given a new block to record information on the blockchain, along with a new cryptocurrency. The main job of miners is transaction confirmation and data packaging.

**Ponzi.** A Ponzi scheme is a traditional investment scam that uses the money of new investors to pay interest and short-term returns to old investors. It is used to create the illusion of making money and then to get more investments. In Ethereum smart contracts, the Ponzi scheme has some new features. Due to the complexity of blockchain-related technology, it is difficult for investors to decipher the specific business logic of an Ethereum smart contract. Generally only a small amount of descriptive information issued by the developer on the smart contract can be used to understand the operation mechanism of the business. This makes Ponzi schemes in smart contracts even more confusing. Many investors believe that the blockchain is tamper-proof, so contracts uploaded into Ethereum will never expire. This has led many investors to believe that a smart contract project that is continuously running and constantly gaining revenue does not have the risk of a Ponzi-like scheme. And mistakenly invested in a Ponzi scheme and ended up losing a lot of money.

**Phish.** While blockchain continues to show vigorous vitality, its own security issues are gradually revealed. Security threats against cryptocurrency applications and various crimes against blockchain platforms are showing a high incidence. In addition to threats such as frequent theft of trading platforms, highlighted vulnerabilities of smart contracts, and crimes committed by using anonymous transactions, phishing frauds committed with the help of blockchain cryptocurrencies are particularly rampant, raising public doubts about the security of blockchain and concerns about its development prospects, and seriously affecting the value storage function of cryptocurrencies.

As for other types of accounts, among them, ICO wallet is a wallet where Token Sale proceeds are/were being stored. Token contract is the address of a smart contract with tokens. Investments are made by large holders of ETH, who usually get in early in the ICO.

## B EXPERIMENTAL SETTING

### B.1 EVALUATION METRICS

In the experiment, four evaluation metrics are chosen to assess the performance of different methods in terms of Ethereum address identification: Macro-Precision, Macro-Recall, Macro-F1, and Micro-F1.

TN (True Negative) represents the number of true negatives for each class.

TP (True Positive) represents the number of true positives for each class.

FN (False Negative) represents the number of false negatives for each class.

FP (False Positive) represents the number of false positives for each class.

$$\text{Macro-Precision} = \frac{1}{n} \sum_{i=1}^{n} \frac{\text{TP}_i}{\text{TP}_i + \text{FP}_i}$$

$$\text{Macro-Recall} = \frac{1}{n} \sum_{i=1}^{n} \frac{\text{TP}_i}{\text{TP}_i + \text{FN}_i}$$

$$\text{Macro-F1} = \frac{1}{n} \sum_{i=1}^{n} \frac{2 \cdot \text{Precision}_i \cdot \text{Recall}_i}{\text{Precision}_i + \text{Recall}_i}$$

$$\text{Micro-Precision} = \frac{\sum_{i=1}^{n} \text{TP}_i}{\sum_{i=1}^{n} \text{TP}_i + \sum_{i=1}^{n} \text{FP}_i}$$

$$\text{Micro-Recall} = \frac{\sum_{i=1}^{n} \text{TP}_i}{\sum_{i=1}^{n} \text{TP}_i + \sum_{i=1}^{n} \text{FN}_i}$$

$$\text{Micro-F1} = \frac{2 \cdot \text{Micro-Precision} \cdot \text{Micro-Recall}}{\text{Micro-Precision} + \text{Micro-Recall}}$$

### B.2 DATASET ANALYSIS

This paper analyzed the statistical values of the node characteristics of different classes of Ethereum accounts. Fig. 2 shows some of the feature *mean* data for the account features. Fig.3 shows the data of the degree distribution features of the account. From this, the following conclusions can be drawn:

(1) Exchanges are an important part of the Ethereum ecosystem. At present, a large number of cryptocurrency transactions are completed through exchanges, maintaining financial connections with a large number of users. As can see from the table that the vast majority of characteristics of exchange accounts are in the top two. In particular, the characteristics of VTS, max_VTR, TVR, URA, ERC20_max_VTS, ERC20_min_VTR and other types of nodes are significantly different from other types of nodes, and transactions occur frequently on exchanges.

(2) As you can see by avg_TIS and avg_RI, ICO wallet accounts are traded less frequently than other accounts. The max_VTS and max_VTR features indicate that there are large-scale transaction behaviors in ICO wallets account transactions. According to the characteristics of TEB, it can be seen that there is a large amount of ether in the ICO wallets account.

(3) For miner accounts, the block reward needs to be transferred to the participant's account as a reward. It can be seen from the statistics that its NTS and USA features are relatively large, which confirms the characteristics of miners' accounts.

(4) Fraud accounts such as Phish and Ponzi have smaller amounts and transaction counts, suggesting that victims of Ethereum phishing and scams have less to lose. Compared with Ponzi and Investment accounts, Token contract accounts generally have larger eigenvalues, which means that Token contract accounts trade more frequently.

(5) From the Fig.3, it can be see that there are some differences in the degree distribution of different types of nodes, especially the degree of Phish and pozi types of nodes is generally smaller than that of other accounts.