# OpenReview forum: "FE-GNN: Feature Enhanced Graph Neural Networks for Account Classification in Ethereum"
_ICLR.cc/2025/Conference — Submitted to ICLR 2025_

### Official Review · Reviewer_sWHP · 2024-11-02

**Soundness:** 1
**Presentation:** 2
**Contribution:** 1
**Rating:** 3
**Confidence:** 4

**Summary:**

The paper explores Ethereum network analysis by combining GCN and GAT models to achieve node representation. It introduces a newly constructed large-scale heterogeneous Ethereum dataset, including 1,124,130 nodes and 3,752,659 edges. The paper discusses the methodology for node classification across multiple categories and provides analyses of label behavior in appendix.

**Strengths:**

1. The author provides detailed explanations and analyses of label behavior in the appendix.
2. The author constructed a large-scale dataset. Although it is not publicly available, the effort is commendable.

**Weaknesses:**

1. Lack of Methodological Innovation: The method mainly combines existing GCN and GAT models to achieve node representation, without proposing any novel methods.
2. Insufficient Experimental Comparisons: The author dedicates significant space in Table 4 to comparisons with fundamental or classical methods such as LR, DeepWalk, and GCN. Only two Ethereum analysis methods are included for comparison, but there is a lack of comparisons with other SOTA (state-of-the-art) Ethereum analysis methods, such as REF 1-4.
3. Logical Gaps in Writing: Many parts lack analysis, leaving the reader confused. For example, in lines 269-272, the author states that there are no existing meta-path related experiments on the Ethereum dataset, and hence proposes a specific process in Fig. 1(a). However, there is no discussion on the differences between the specific process proposed and existing processes for other heterogeneous graphs, why existing methods cannot be directly used, what issues they have, why those issues pose challenges, and what specific solutions the author proposes. The internal mechanism of the proposed method is also unexplained. Similar issues are prevalent throughout the paper.
4. Unclear Explanations in Key Areas: The dataset constructed is a contribution, but essential details are missing. What is the timeframe of the transactions in the dataset? Are they from recent transactions in 2024 or from earlier years? The author does not analyze or consider potential evolutions and changes in transaction behaviors across different years. Since the dataset is highlighted as a key contribution, it is recommended to make it publicly available.
5. Numerous Minor Writing Issues: For example, commas are needed after formulas 10, 11, and 12. Similar minor issues are present throughout the text.



REF1. Yang, J., Yu, W., Wu, J., Lin, D., Wu, Z., & Zheng, Z. (2024). 2DynEthNet: A Two-Dimensional Streaming Framework for Ethereum Phishing Scam Detection. IEEE Transactions on Information Forensics and Security.

REF2. Lin, D., Wu, J., Fu, Q., Zheng, Z., & Chen, T. (2024). RiskProp: Account risk rating on Ethereum via de-anonymous score and network propagation. IEEE Transactions on Dependable and Secure Computing.

REF3. Liu, J., Chen, J., Wu, J., Wu, Z., Fang, J., & Zheng, Z. (2024). Fishing for Fraudsters: Uncovering Ethereum Phishing Gangs With Blockchain Data. IEEE Transactions on Information Forensics and Security.

REF4. Wu, J., Lin, D., Fu, Q., Yang, S., Chen, T., Zheng, Z., & Song, B. (2023). Towards Understanding Asset Flows in Crypto Money Laundering Through the Lenses of Ethereum Heists. IEEE Transactions on Information Forensics and Security.

**Questions:**

1. Could the author explain why the collected dataset was divided into $D_1,D_2, D_3$​ as shown in Table 3? Why not conduct experiments on the entire dataset?
2. The effort in collecting the Ethereum dataset, which includes 1,124,130 nodes and 3,752,659 edges, is commendable. However, necessary clarifications about the dataset are needed. For example, what is the timeframe of the transactions included in the dataset? Does it cover all transactions during that period? Was any preprocessing done? According to the descriptions in the introduction and methods section, the constructed graph dataset is a heterogeneous graph. What are the proportions of each node category in the 1,124,130 nodes?
3. Could the author further analyze the fairness of the experiments in Table 4? According to the descriptions, the author considers an eight-class problem (as shown in Table 2). However, as per the experimental section, $T^2A2vec$ and HNRL are phishing node detection methods, i.e., binary classification. How were the experimental settings aligned in this context? If $T^2A2vec$ and HNRL are primarily focused on detecting phishing nodes, is it fair to compare their effectiveness in identifying Exchange, ICO Wallets, and the other seven categories?

---

### Official Review · Reviewer_bagd · 2024-11-02

**Soundness:** 2
**Presentation:** 2
**Contribution:** 2
**Rating:** 3
**Confidence:** 5

**Summary:**

The paper presents Feature Enhanced Graph Neural Networks, a method for identifying cryptocurrency addresses in Ethereum to improve transaction analysis. By building a transaction graph and using graph learning techniques, FE-GNN embeds blockchain addresses,  and the embeddings demonstrate superior performance compared to traditional methods in its experimental evaluations.

**Strengths:**

- Blockchain data analytics is an interesting and novel research area.
- Experiments use related models and compare the results. This is the strongest part of the article.

**Weaknesses:**

I see two main weaknesses:

- The evaluation of FE-GNN mainly depends on a single Ethereum dataset. For more comprehensive validation, results should be tested on additional datasets, such as the phishing dataset https://www.kaggle.com/datasets/xblock/ethereum-phishing-transaction-network. The dataset is not ours.
- Even if the dataset limitation is addressed, the approach is iterative and lacks novelty. Using GNNs and attention mechanisms is standard practice. The authors should probably open a new novelty path to publish the article at a premier conference.

Additional concerns include:

- While the paper mentions 1,124,130 nodes and 3,752,659 edges, it doesn’t specify the time period over which this data was collected.
- The evaluation overlooks the temporal aspect of transactions, which is crucial for capturing changes in user behavior over time, especially given how quickly trends can shift in the cryptocurrency space.
- The paper doesn’t discuss the computational costs associated with the FE-GNN method, but the layered use of convolutions and self-attention mechanisms is likely to demand significant computational resources.
- There is no ablation study to assess the contribution of each component or step in the FE-GNN algorithm. Without this, it’s unclear which parts of the method are driving improvements, making it hard to pinpoint optimal configurations or areas for improvement.

**Questions:**

- What is the time complexity?
- What are the studied time periods?
- What are the results of a temporal prediction task? You could train a model with data until t and predict crime nodes at t+1. Such an experiment would prove the value of the model.

---

### Official Review · Reviewer_5Tzr · 2024-11-03

**Soundness:** 3
**Presentation:** 3
**Contribution:** 3
**Rating:** 6
**Confidence:** 4

**Summary:**

The paper proposes a novel blockchain address identity identification method called Feature-Enhanced Graph Neural Network (FE-GNN) for improving Ethereum account classification. It constructs a transaction graph with over 1 million nodes and 3.7 million edges from Ethereum transactions and applies a GCN (Graph Convolutional Network) and GAT (Graph Attention Network) hybrid model to enhance node representation. The model is evaluated using an extensive dataset containing over a million nodes and achieves notable performance gains over traditional methods for identifying specific Ethereum account types.

**Strengths:**

The use of a hybrid GCN-GAT model and focus on feature enhancement is a creative solution that improves classification accuracy by addressing node heterogeneity. The experiments are rigorously conducted, with robust comparisons to existing models. The division of the dataset and metrics used are appropriate, though more detail on feature selection could clarify which elements were most impactful. The methodology is generally clear, but certain technical terms and detailed GNN layer operations may require additional clarification for broad accessibility. More visual aids might benefit understanding. Given the increasing importance of transaction monitoring in Ethereum, this work could have a positive impact on blockchain research, especially in enhancing identity tagging.

**Weaknesses:**

1. The paper lacks detailed justification for specific hyperparameter values (e.g., convolutional layers, and self-attention heads). A sensitivity analysis could enhance reproducibility and model tuning.
2. Although the dataset is thoroughly described, more information on how validation and test splits influence model training could strengthen reproducibility.
3. Adding insights into why certain models were selected would clarify the robustness of the results.
4. While feature extraction is thoroughly described, including a breakdown of features’ contributions to classification accuracy would add depth to the evaluation.

**Questions:**

1. Could the authors explain if and how the model's performance might vary with other blockchains?
2. Is there any potential for applying this model in real-time transaction monitoring or identifying emerging account types?
3. Would varying the types of features collected (e.g., smart contract interactions) impact the results?
2. Could the authors provide further justification or a sensitivity analysis for the specific configurations of the convolutional and self-attention layers?
3. Could the authors explore which extracted features most significantly contributed to the FE-GNN’s classification performance?
4. Have the authors considered training with imbalanced classes (e.g., phishing, exchanges) to reflect real-world scenarios?

---

### Official Review · Reviewer_C77B · 2024-11-03

**Soundness:** 3
**Presentation:** 3
**Contribution:** 2
**Rating:** 5
**Confidence:** 5

**Summary:**

The paper proposes a modified graph neural network architecture tailored for identifying node types in Ethereum graphs.

**Strengths:**

+ The paper addresses an important problem

+ The paper is easily understandable

+ Experiments done on a real ethereum dataset

**Weaknesses:**

- The proposed approach seems like incremental improvement over existing GNN approaches

- The results do not provide error bars. It is not clear to me whether the improvement shown is statistically significant.

- Some important baseline GNN models such as Graphsage has not been used. Given the performance of GAT, it is not clear to me whether the other existing GNN approaches perform even better.

- Also some of the node types are easy to classify, e.g., token contracts, by looking at the node feature so it was not clear to me where the improved performance came.

**Questions:**

Please see the above week points.

---

### Meta-Review · Area_Chair_aJTZ · 2024-12-19

**Metareview:**

The paper proposes a novel blockchain address identity identification method called Feature-Enhanced Graph Neural Network (FE-GNN) for improving Ethereum account classification, which involves transaction graphs with over 1 million nodes and 3.7 million edges. The evaluation results demonstrate their methods' effectiveness against traditional methods.


Strengthens:

1. The problem studied in this paper is an important problem.


Weaknesses:

1. The proposed methods don't show much technique novelties when applying GCN or GAT to blockchains. The novelty is limited.

2. The evaluation settings are limited.

All reviewers propose the concerns on the paper's novelty and empirical settings but without any responses during the rebuttal phase. Although the problem studied in this paper is interesting and important, this paper is still immature for the ICLR.

**Additional Comments On Reviewer Discussion:**

All reviewers propose the concerns on the paper's novelty and empirical settings but without any responses during the rebuttal phase.

---

### Decision · Program_Chairs · 2025-01-22

Reject